# The Impact of Rest Intervals on the Force–Velocity Relationship Obtained During the Keiser’s 10-Repetition Leg Press Test

**DOI:** 10.3390/jfmk10010031

**Published:** 2025-01-14

**Authors:** John Magne Kalhovde, Christian M. M. Johannessen, Sigmund B. Aasen, Espen Tønnessen

**Affiliations:** 1School of Health Sciences, Kristiania University College, 0153 Oslo, Norway; johnmagne.kalhovde@kristiania.no (J.M.K.); sigmund.apold-aasen@kristiania.no (S.B.A.); 2Department of Education and Sports Science, University of Stavanger, 4036 Stavanger, Norway; magnus.johannessen@skole.rogfk.no

**Keywords:** physical testing, force–velocity test, power, mobilizing, rest intervals, recovery

## Abstract

**Background:** The Keiser 10-rep leg press test protocol employs short inter-repetition rest intervals (2–38 s), raising concerns as to whether athletes perform optimally. **Objectives:** The aim of this study was to compare the results of the standard Keiser protocol with an identical protocol modified to include a significantly longer inter-repetition rest intervals and to evaluate whether these effects differed between men and women. **Methods:** A total of 30 athletes (age 17.1 ± 0.9 years, height 177.8 ± 8.8 cm, and body mass 71.9 ± 11.3 kg) performed three separate tests (total of ~900 repetitions) in a Keiser A420 leg press machine, separated by 4 to 8 days. Test day 1 included a 1RM test followed by a 10-repetition force–velocity test with a standard rest intervals. Test days 2 and 3 involved the same test, with either standard short rest intervals or fixed 3 min inter-repetition rest intervals. **Results:** Increasing inter-repetition rest intervals significantly improved *V*_0_ and *P*_max_ for men and *V*_0_ and *FV*_slope_ for women. The benefits of longer rest were notably more pronounced in men, particularly at light to moderate loads, where standard Keiser rest intervals are short (2–9 s). However, extending rest intervals beyond approximately 30 s, as applied at higher loads, provided no additional advantages. **Conclusions:** Longer rest intervals improved force–velocity metrics more in men than women, with the effect being most pronounced at light to moderate loads where standard protocols utilize short rest intervals (2–9 s). These findings suggest that women recover faster than men under these conditions. However, extending rest beyond 30 s at higher loads provides no additional benefits and is counterproductive at maximal loads.

## 1. Introduction

Force–velocity (FV) profiling, and specifically the Keiser 10-rep test, has seen a significant rise in popularity over the last decades [1,2]. Due to the fixed seating position and back support, machine exercises such as this are considered safe, less technically demanding, and require less familiarization compared to more traditional free-weight exercises [3]. In sports requiring significant strength, power, and speed, FV profiling has become an essential tool for assessing performance, monitoring training adaptations, and providing coaches with valuable insights to optimize training programs for athletes [4,5,6,7]. The concept of FV profiling is grounded in the fundamental properties of skeletal muscle, wherein an inverse relationship exists between force and velocity [8,9].

The FV relationship in multi-joint movements exhibits a linear pattern, contrasting with the hyperbolic relationship seen in isolated muscles or single-joint movements [8]. This linearity allows the calculation of a linear regression line and the extrapolation of its endpoints, identifying four key metrics: the theoretical maximum force (*F*_0_), representing the force at zero velocity, and the theoretical maximum velocity (*V*_0_), representing the velocity at zero force. Additionally, the theoretical maximum power is calculated using the formula (*F*_0_·*V*_0_)/4, and the slope of the FV profile (*FV*_slope_) is determined as *F*_0_/*V*_0_ [6,10].

For assessing lower body musculature, force–velocity (FV) profiling has most commonly been determined using vertical jumps with incremental loads [11,12]. However, this method has limitations, including the technical difficulty of jumping with heavy loads near *F*_0_ and the inability to measure velocities close to *V*_0_ due to body weight being the lightest possible load [6]. The pneumatic resistance-based Keiser leg press, widely used in sports and research facilities globally, offers several advantages over traditional weight-based exercises [13]. One of the main advantages with pneumatic-based resistance is that it allows for more precise assessment near the *F*_0_ and V_0_ intercepts since the effects of inertia and body weight are minimized. Obtaining measurements closer to *F*_0_ and *V*_0_ has been shown to produce more reliable results [6,14]. The Keiser A420 leg press is equipped with a pre-installed, standardized protocol for force–velocity profiling based on a series of 10 repetitions at different loads. The test requires an input of an estimated 1RM value, from which it calculates appropriate loads for each repetition necessary to obtain a full FV profile [15].

To achieve a valid test result that can be used to assess development in high-level athletes, it is crucial that the test accurately reflects the performance capacity [16]. In the context of FV profiling, using appropriate inter-repetition rest intervals is crucial for achieving full recovery, which in turn facilitates consistent and optimal performance results [17,18,19]. In the Keiser A420 10-rep test, the inter-repetition rest intervals are pre-determined, ranging from 2 s between the two lightest loads (reps 1 and 2) to 38 s between the two heaviest loads (reps 9 and 10), which diverge from widely accepted recommendations for maximal-effort strength, speed, and power testing [17,20,21,22]. For one-repetition maximum (1RM) testing, rest intervals of 2 to 5 min are commonly recommended [20,23]. In assessments of explosive strength and sprint, rest intervals of 1 to 3 min have been suggested as optimal [22]. These rest intervals are reference ranges and may need to be adjusted based on gender, age, sport, and training status [24,25,26]. Previous studies have indicated that strong individuals with a significant amount of muscle mass may require longer rest periods than weaker individuals with less muscle mass [27]. A few studies suggest that fatigue during isometric strength testing occurs earlier in men compared to women [28,29,30]. However, these sex differences depend on the type of contraction [29], intensity [31], and muscle group [28].

To our knowledge, no previous studies have examined the effect of increased inter-repetition rest intervals during force–velocity profiling using the Keiser A420 leg press or explored potential gender-specific differences. The aim of this study was to compare the results of the standard Keiser protocol with a modified protocol that included significantly longer inter-repetition rest intervals, focusing on the main outcome variables (*V*_0_, *F*_0_, *P*_max_, and *FV*_slope_) and evaluating whether these effects differed between men and women.

## 2. Materials and Methods

### 2.1. Participants

A total of 30 participants, all enrolled in a sports program at the same high school in Norway, were recruited for this study. The inclusion criteria for recruitment into the study required that participants were aged between 16 and 19 years, actively training and competing at the national level in their respective sport [32], injury-free, and available to complete the planned force–velocity tests. During the recruitment process, we also aimed for gender balance to address the research questions in the study as effectively as possible. Their age, height, and body mass (Mean ± SD) were 17.1 ± 0.9 years, 177.8 ± 9.0 cm, and 71.9 ± 12.1 kg. The study included both male (*n* = 14; 16.9 ± 0.9 years; 185.3 ± 6.6 cm; 77.7 ± 12.1 kg) and female (*n* = 16; 17.1 ± 0.8 years; 171.3 ± 4.7 cm; 66.4 ± 8.1 kg) athletes. Fifteen of the participants were handball players, and the remaining 15 participants were endurance athletes competing in track (middle distance and long distance, *n* = 6), orienteering (*n* = 1), cycling (*n* = 2), rowing (*n* = 5), and swimming (*n* = 1). All participants competed at a national junior level in their respective sport (Table 1).

### 2.2. Experimental Approach to the Problem

All participants completed three days of testing (Table 2), separated by 4 to 8 days to allow for adequate physical recovery. All tests were performed on a Keiser A420 pneumatic leg press machine (Keiser Corporation, Fesno, CA, USA) at the Norwegian Olympic Federation test center in Stavanger, Norway. The A420 software from Keiser (version 9.4.5) was used in the testing of the athletes. All tests were conducted by the same test leader, who has significant experience in force–velocity testing. On Test Day 1, participants underwent a 1RM (one-repetition maximum) test protocol, followed by a familiarization session using the Keiser 10-repetition force–velocity (FV) profiling test. Standard rest intervals of 2–38 s were implemented between repetitions during the familiarization session. The recovery time between the two tests was 5 min. During test day 2 and 3, the participants performed either the Keiser 10-repetition FV-profiling test with standard (Short) incremental rest intervals of 2–38 s or fixed manually timed 3 min rest intervals (Long) between each repetition. Participants were randomized into two groups, ensuring a balanced distribution of gender and sport background in each group. Differentiating the test order aimed to minimize the potential influence of test familiarization on the results.

The study was approved by the Norwegian Center for Research Data (SIKT reference number: 339372). The study was performed in agreement with the Declaration of Helsinki, and all participants gave their consent to participate in it.

### 2.3. Procedures and Warm-Up

On test day 1, body weight and height were measured using a Seca 877 floor scale and a Seca 222 mechanical telescopic measuring rod (Seca GmbH & Co. KG, Hamburg, Germany). All subjects performed a standardized warm-up prior to each test session. This included a general warm-up consisting of 5 min of ergometer rowing (pace 2–2:30 min/500 m), followed by a specific warm-up on the Keiser leg press, consisting of 1 × 3 repetitions at 20%, 40%, and 60% of estimated 1RM, with a 1 min rest interval between each load level. In advance, all participants received written instructions to prepare themselves for each test sessions as they would for a competition with regard to diet, fluid intake, and rest. They were also instructed to avoid any strenuous exercise 24 h prior to testing. The equipment, test leader, and location were identical for all tests.

### 2.4. Strength Testing Protocols

For both the 1RM and Keiser’s 10-rep tests, each participant’s seat on the Keiser A420 leg press was adjusted to achieve a nearly vertical femur, with the position recorded and consistently used across all tests. Feet were positioned with the heels flush at the lower end of the foot pedal to standardize testing across participants.

Maximum strength was assessed using Earle’s 1RM testing protocol [33,34]. Following a warm-up, participants rested for 2 min before doing 2–3 repetitions at a submaximal load equivalent to approximately 80% of their predicted 1RM. The load was then progressively increased, with 3 min rest intervals between attempts, until 1RM was achieved. All participants successfully determined their 1RM within 3–5 attempts.

All participants underwent Keiser’s 10-rep test protocol on three separate occasions (Table 2). The value from the 1RM test was entered, and based on this, the Keiser A420 software (version 9.4.5) calculated and dictated individual test loads. Participants were instructed to execute each repetition with maximum effort during the concentric phase, starting from the predetermined pedal position ensuring concentric-only actions without countermovement. The eccentric phase was not recorded. Rep-by-rep values for loads relative to 1RM and inter-repetition rest intervalss under the two test conditions is shown in Table 3.

### 2.5. Statistical Analyses

A repeated measures multivariate analysis of variance (RM-MANOVA) was conducted to examine the overall effect of experimental conditions on the four dependent variables (*V*_0_, *F*_0_, *P*_max_, and *FV*_slope_) while accounting for within-subject variability and the correlations among the dependent variables. The analysis included one within-subject factor, the rest interval, with two levels: “Short” and “Long”. The RM-MANOVA tested the main effect of the rest intervals and the interaction between the two levels of conditions (Short and Long) and the dependent variables to determine whether the condition’s effect varied across the different measures.

Before conducting the RM-MANOVA, data were restructured, and key assumptions were tested and confirmed within acceptable limits. These included normality (Shapiro–Wilk test), the homogeneity of covariance matrices (Box’s M test), and sphericity (Mauchly’s test). Correlations among the dependent variables were also examined, confirming that multicollinearity was not a concern.

Follow-up univariate analyses were conducted for each dependent variable, based on gender, to identify specific effects. A significance level of α = 0.05 was applied to all statistical analyses. Bonferroni correction was applied to control for the Type I error across multiple comparisons. Effect sizes, including partial eta squared (*η*^2^) and Cohen’s d, are reported to indicate the magnitude of the observed differences. Partial eta squared values were interpreted as small (*η*^2^ = 0.01), medium (*η*^2^ = 0.06), and large (*η*^2^ = 0.14), while Cohen’s d values were interpreted as small (*d* = 0.2), medium (*d* = 0.5), and large (*d* = 0.8). All statistical analyses were conducted using SPSS (version 28, IBM Corp., Armonk, NY, USA). GraphPad Prism (version 10.4.0, GraphPad Software, San Diego, CA, USA) was used for the data visualization and graphical representation of the results.

## 3. Results

The initial analysis (RM-MANOVA) showed no significant difference of inter-repetition rest interval length when considering all dependent variables together (*F*1,29 = 0.329, *p* = 0.571, partial *η*^2^ = 0.011). However, a significant interaction effect was found between rest interval intervals and dependent variables (*F*3,27 = 8.969, *p* < 0.001, partial *η*^2^ = 0.499), suggesting that the effect of rest interval length varied across the dependent variables. Furthermore, a significant interaction effect was found between gender, rest interval length, and dependent variables (*F*3,24 = 3.846, *p* = 0.022, partial *η*^2^ = 0.325).

Follow-up univariate analyses revealed gender-specific responses to inter-repetition rest intervals. Among women, *V*_0_ increased significantly following long rest intervals compared to short rest intervals (Mean ± SD: 2.28 ± 0.26 m/s^2^ vs. 2.21 ± 0.26 m/s^2^; *p* = 0.015), representing a 3.19% improvement with a small effect size (*d* = 0.26). For men, the increase in *V*_0_ was even more pronounced (Mean ± SD: 2.50 ± 0.24 m/s^2^ vs. 2.31 ± 0.28 m/s^2^; *p* = 0.003), corresponding to an 8.50% improvement with a large effect size (*d* = 0.71). These findings suggest that extended recovery optimizes velocity-dominant performance in both sexes, with a greater impact in men. Maximum force (*F*_0_) exhibited no significant differences between rest intervals for either women (Mean ± SD: 1730.6 ± 267.2 N vs. 1758.4 ± 279.2 N; *p* = 0.282) or men (Mean ± SD: 2394.1 ± 524.4 N vs. 2453.1 ± 494.6 N; *p* = 0.159). The mean differences (−1.45% and −2.75%, respectively) were minimal, with small effect sizes (*d* = 0.10 and *d* = 0.12). Peak power (*P*_max_) increased following long rest intervals in men, with a significant improvement (Mean ± SD: 1597.4 ± 442.8 W vs. 1505.4 ± 388.2 W; *p* = 0.006), representing a 5.74% increase and a small effect size (*d* = 0.22). In women, the difference in *P*_max_ was not significant (Mean ± SD: 1033.6 ± 216.1 W vs. 1013.8 ± 206.1 W; *p* = 0.280), with a mean increase of 1.90% and a small effect size (*d* = 0.09). The force–velocity slope (*FV*_slope_) became steeper following long rest intervals in both sexes. For women, the difference was significant (Mean ± SD: −0.0131 ± 0.0022 N·m/s vs. −0.0125 ± 0.0022 N·m/s; *p* = 0.023), with a 4.80% increase and a small effect size (*d* = 0.25). For men, the change approached significance (Mean ± SD: −0.0105 ± 0.0021 N·m/s vs. −0.0094 ± 0.0016 N·m/s; *p* = 0.057), corresponding to an 11.91% increase with a moderate effect size (*d* = 0.58).

The individual test results for each metric, shown in Figure 1a–d, provide a detailed visualization of participant-level changes and between-subject variability. Figure 1a highlights consistently higher *V*_0_ values with long rest intervals across most participants, with a more pronounced increase observed in men compared to women. In contrast, Figure 1b demonstrates similar *F*_0_ values regardless of rest condition, indicating the minimal impact of rest interval length on force output. Figure 1c illustrates a clear trend toward higher *P*_max_ with longer rest intervals for both genders; however, the difference is statistically significant only for men. Finally, Figure 1d shows an overall trend of steeper *FV*_slope_ with longer rest intervals. The change is statistically significant for women, while men exhibited a near-significant trend (approaching, but not reaching, the alpha-level of 0.05). These figures illustrate the range of individual adaptations to different rest interval protocols, complementing the statistical findings reported in Table 4.

Figure 2 presents the relative changes in performance metrics (velocity, force, and power) from short to long rest intervals during the Keiser 10-rep test. For submaximal repetitions (1–9), men produced notably higher velocities improvements at low to moderate loads, with relative changes ranging from 3.8 to 7.4%, compared to women, who ranged from −0.2% to 1.6%. Except for repetition 1 and 2 for women, force output changes were similar between genders, showing slightly higher but trivial differences across repetitions. Relative changes in power, being a product of force and velocity, showed the greatest changes. As for velocity, the largest relative increase in power for men were observed at repetitions 1 through 6, ranging from 5.4% to 10.1%, while women showed smaller relative increases, ranging from 2.0% to 5.1% over the same repetition range. For maximal-effort attempts (repetition 10), both men and women performed notably worse with longer rest intervals, with considerable variability. Men reduced their velocity by 6.6% and power by 6.3%, whereas women showed sharper declines, with velocity reduced by 15.1% and power by 12.5%.

## 4. Discussion

This study aimed to investigate if longer inter-repetition rest intervals influence force–velocity (FV) metrics—specifically *V*_0_, *F*_0_, *P*_max_, and *FV*_slope_—during the Keiser A420 leg press 10-rep test. A significant interaction revealed that the impact of inter-repetition rest intervals varied across the FV metrics and with gender-specific differences. Follow-up analyses demonstrated significant improvements in *V*_0_ and *P*_max_ for men and *V*_0_ and *FV*_slope_ for women. Individual repetition analysis revealed a load specific effect, indicating that the very short rest intervals (2–9 s) used in the Keiser 10-rep test at low and moderate loads impair optimal performance. This effect was more pronounced in men, who benefitted significantly more from longer rest intervals, achieving greater relative increases in velocity and power compared to women.

Accurate assessments of metrics such as *V*_0_, *F*_0_, *P*_max_, and *FV*_slope_ are critical for tailoring athlete training programs [5,13]. When assessing athletic performance, even subtle improvements can be meaningful if they represent a genuine enhancement in an athlete’s capabilities rather than variability caused by testing protocols, instrumentation, or fluctuations in athlete preparedness on a given day. In sports testing, the smallest worthwhile change (SWC) is frequently calculated as 0.2 × the baseline standard deviation (SD), as suggested by Cohen [35,36]. Applying this threshold to our data, *V*_0_, *P*_max_, and *FV*_slope_ exceed the SWC for men, while *V*_0_ and *FV*_slope_ exceed the threshold for women. However, for elite athletes, even smaller relative changes might be considered worthwhile, given the minimal margins often separating competitors at the highest level [37,38].

Another important consideration when interpreting test performance data is the reliability of the test itself. Reliability ensures that observed differences are not the result of a random measurement error. In this context, the typical error (TE), a widely recognized measure of reliability, quantifies the random variation inherent in repeated measurements [36]. Although the TE was not explicitly reported in this study, all differences in FV variables, except for *F*_0_, exceeded TE thresholds. This indicates that the performance changes due to longer rest intervals are unlikely due to random variability, thereby underscoring the practical significance of our findings. Together, the evaluation of SWC and TE strengthens the confidence that the reported improvements in this study reflect real and meaningful changes in performance rather than measurement noise.

Our findings suggest that men require more time to recover from maximal-effort single repetitions at low and moderate loads compared to women. Sex differences in fatigue are complex and shown to be task-dependent [39]. While some studies report no significant differences between men and women [40], others indicate that men are often more fatigable than women during sustained or intermittent isometric exercise, and it is suggested that this greater fatigability in men is likely due to higher levels of central fatigue [39,41]. Our results align with these observations, demonstrating that, despite involving single repetitions at loads as low as 15% of 1RM, each repetition requires maximal effort and elicits the highest possible rate of force development (RFD), demanding intense bursts of action potentials from the nervous system. For men, the short inter-repetition rest intervals (2–9 s) used during repetitions 1 through 6 may not provide sufficient time for the nervous system to fully recover and prepare for another maximal-effort action potential discharge. In contrast, these short rest intervals appear more adequate for women, suggesting faster recovery dynamics in their neuromuscular systems. Other explanations for the benefits of longer rest intervals might lie within the muscle cells’ energy delivery system as high intensity efforts rely on fast-acting glycolytic energy systems. Depleting PCr stores and accumulating metabolic byproducts, such as H^+^ ions, have previously been shown to impair muscle function [42]. Extending rest intervals beyond 2–9 s would thus increase muscle cells’ ability for sufficient PCr replenishment and pH stabilization, potentially enabling higher force and velocity outputs [19,23].

In contrast to the results observed at light and moderate loads, and somewhat unexpectedly, extending rest intervals at loads above 80% of 1RM (repetitions 8–10) did not improve performance. One possible explanation for this finding might lie in the fact that resting for as long as 3 min might be counterproductive. Extending inter-repetition rest intervals longer than needed might cause participants to lose focus and the optimal level of arousal, thereby reducing the physiological basis to optimally perform maximal-effort attempts. In fact, this sensation was reported to the test leader by several of the athletes during and after testing. In accordance with this hypothesis, a study found no significant benefits of additional rest beyond one minute when performing FV profiling using a loaded squat jump [19]. As shorter rest intervals were not tested in their study, and considering our findings, it is plausible that a threshold exists somewhere around 30–60 s, beyond which no further improvement in performance can be achieved. Extending rest intervals beyond this time frame may negatively impact maximal-effort test results during single repetitions, such as in force–velocity profiling.

The substantial inter-subject variation observed at high loads, particularly during repetition 10, likely results from a combination of factors. Firstly, the final repetition in the Keiser 10-rep test represents the participant’s 1RM. Notably, 6 out of the 30 participants were unable to complete a full concentric repetition at their predetermined 1RM (established during test day one) on one or both of the subsequent test days. Despite instructions to prepare consistently, as they would for competition, individual differences in readiness and day-to-day variations are inevitable. Additionally, we believe that excessively long rest intervals may have contributed to the increased variance observed at higher loads, likely due to fatigue from prolonged intervals and a loss of optimal arousal levels, as previously mentioned. While rest protocols were counterbalanced for half of the participants to reduce familiarization and learning effects, the load sequence remained consistent across all participants. Future research should explore reversing the load sequence, starting with the highest loads, to assess whether prolonged test intervals and potential psychological fatigue are specifically related to the load or the overall testing time.

A strength of the present study is the use of validated testing equipment and a standardized test protocol that has been validated in previous studies [13,15]. Additionally, the study included participants who train and compete at an elite level [32]. However, the study has limitations regarding experience with strength testing as the participants have a relatively low strength-related training age. To achieve greater statistical power, we could have increased the number of participants [43]; however, recruiting higher numbers of elite athletes presents significant challenges. Furthermore, as we did not collect psychological and physiological data at the nerve and muscle levels, we were unable to explain or understand the mechanisms underlying our findings. Future studies should investigate whether changes in rest intervals have different effects on men and women and how any potential differences can be explained based on psychological and physiological factors [19,44]. The main goal for future research should, therefore, focus on optimizing load-specific rest interval lengths, without compromising test efficiency.

From a methodological standpoint, our findings emphasize the importance of aligning FV-profiling protocols with physiological and psychological recovery principles. Furthermore, while 3 min rest intervals were used in this study, reducing unnecessarily long rest intervals—and thereby shortening the total test intervals—would be advantageous from a practical perspective. Given that tests are often conducted on larger groups or teams, logistical efficiency, cost-effectiveness, and time management are important considerations. Future research should, therefore, aim to identify the optimal recovery time required to balance these practical constraints with an accurate performance assessment.

## 5. Conclusions

In conclusion, this study highlights the need to re-evaluate inter-repetition rest interval lengths in FV profiling using the Keiser A420 leg press. Longer rest intervals improved force–velocity metrics for both genders, but more in men than women. The effect of longer rest intervals was most pronounced at light to moderate loads where standard protocols utilize short rest intervals (2–9 s). These findings suggest that women recover faster than men when performing repeated single maximal-effort contractions. However, extending rest beyond 30 s at higher loads provides no additional benefits and seems counterproductive at maximal loads for both genders.

## Figures and Tables

**Figure 1 jfmk-10-00031-f001:**
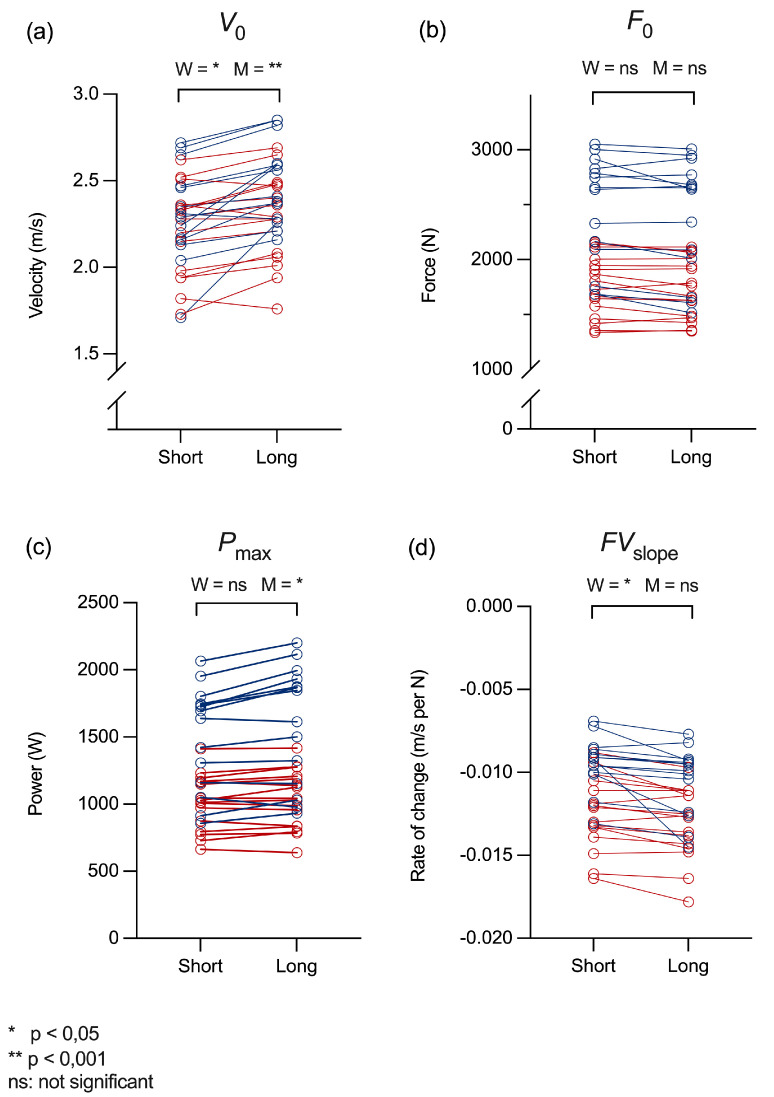
(**a**–**d**) Individual test results (men in blue, women in red) for the Keiser 10-rep test for *V*_0_ (**a**), *F*_0_ (**b**), *P*_max_ (**c**), and *FV*_slope_ (**d**). “Short” refers to standard rest intervals as dictated by Keiser’s 10-rep test protocol (2–38 s), while “Long” represents consistent 3 min rest intervals between each repetition.

**Figure 2 jfmk-10-00031-f002:**
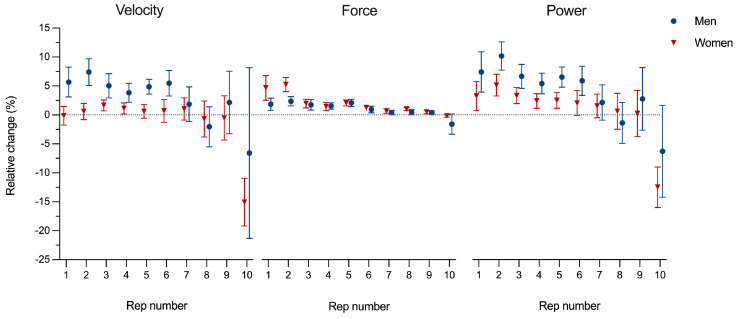
Mean values for men (blue) and women (red) across repetitions, ranging from light (1) to maximal (10) loads, for velocity, force, and power. Error bars represent 95% confidence intervals.

**Table 1 jfmk-10-00031-t001:** Baseline characteristics of the athletes in the study (*n* = 30).

Variable	All Athletes (*n* = 30)		Endurance Athletes (*n* = 15)		Handball Athletes (*n* = 15)	
Sex (*n*) Age (years)	F (16) 17.1 ± 0.8	M (14) 16.9 ± 0.9	F (8) 17.5 ± 0.8	M (7) 16.2 ± 0.4	F (8) 16.9 ± 0.8	M (7) 17.7 ± 0.5
Height (cm)	171.3 ± 4.7	185.3 ± 6.6	171.3 ± 5.5	180.6 ± 4.9	173.0 ± 11.3	189.9 ± 7.6
Weight (kg)	66.4 ± 8.1	71.9 ± 12.1	64.5 ± 10.3	71.9 ± 10.1	74.8 ± 12.7	86.8 ± 5.9

**Table 2 jfmk-10-00031-t002:** The content of the test days.

Group	Test Day 1	Test Day 2	Test Day 3
1	1RM and FV profiling (Short)	FV profiling (Short)	FV profiling (Long)
2	1RM and FV profiling (Short)	FV profiling (Long)	FV profiling (Short)

**Table 3 jfmk-10-00031-t003:** Rep-by-rep values for loads relative to 1RM and inter-repetition rest intervals under the two test conditions: Short (standard) and Long (fixed 3 min).

Repetition:	1	2	3	4	5	6	7	8	9	10
% of 1RM	15	27	35	45	54	63	72	82	91	100
Short rest intervals (s) *	2	5	5	6	9	13	18	26	38	
Long rest intervals (min) **	3	3	3	3	3	3	3	3	3	

* The total recovery time was 2:08 min using short rest intervals. ** The total recovery time was 27:00 min using long rest intervals.

**Table 4 jfmk-10-00031-t004:** Mean values and results from univariate analyses of all FV metrics calculated using the Keiser 10-repetition test, with Bonferroni-corrected *p*-values, for standard short (2–38 s) and long (3 min) inter-repetition rest intervals.

FV-Variable	Mean Value (±SD)		Mean Difference (%)	Effect Size	*p*-Value
	Short	Long		Cohen’s *d*	
**Women:**					
*V*_0_ (m/s^2^)	2.21 (0.26)	2.28 (0.26)	3.19	0.26	0.015
*F*_0_ (N)	1758.4 (279.2)	1730.6 (267.2)	−1.45	0.10	0.282
*P*_max_ (W)	1013.8 (206.1)	1033.6 (216.1)	1.90	0.09	0.280
*FV*_slope_ (N m/s)	−0.0125 (0.0022)	−0.0131 (0.0022)	4.80	0.25	0.023
**Men:**					
*V*_0_ (m/s^2^)	2.31 (0.28)	2.50 (0.24)	8.50	0.71	0.003
*F*_0_ (N)	2453.1 (494.6)	2394.1 (524.4)	−2.75	0.12	0.159
*P*_max_ (W)	1505.4 (388.2)	1597.38 (442.8)	5.74	0.22	0.006
*FV*_slope_ (N m/s)	−0.0094 (0.0016)	−0.0105 (0.0021)	11.91	0.58	0.057

## Data Availability

The datasets generated and/or analyzed during the current study can be obtained from the corresponding author upon reasonable request.

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
