# Peer review of "The Impact of Rest Intervals on the Force–Velocity Relationship Obtained During the Keiser’s 10-Repetition Leg Press Test"

_jfmk, 2025, doi:10.3390/jfmk10010031_

Round 1
Reviewer 1 Report
Comments and Suggestions for Authors
Thanks a lot for the opportunity to score this ms. The work is descritive and results expected. Nevertheless, is well conducted. Overall it adds some info for professionals in the field. A fundamental limitation of the experimental design is that males and females were treated as a single homogeneous group, without accounting for the physiological differences that could influence the results. It is well established that the fatigue effects of exercise can vary significantly between genders. This point should be adequately addressed both in the text and in the statistical section. Why statistics does not include sex as a factor or covariate? this lack should be justified or analysis should be redone and text adequately modified.
Reviewer 2 Report
Comments and Suggestions for Authors
Dear friends, considerations for the study that aimed to "investigate the effect of using short (2-38s) or long (3min) rest intervals on maximum strength (F0), speed (V0), power (Pmax) and the slope of the FV profile (FVslope) in the Keiser 10-repetition leg press test".
Abstract
1. I suggest adjusting the "objective" of the study presented in this section with the "objective" presented at the end of the Introduction section. They should be the same. In the Abstract, the authors talk about "comparing" in the Introduction; it reads "investigating". Also, the statements of the two objectives, although they seek the same results, are different!
Introduction
1. With the exception of the objective that must be adjusted with the Abstract, the other information was well constructed.
Methodology
1. Table 1 (Baseline characteristics of the athletes in the study (n = 30) should be presented in the Results section;
2. I suggest creating a section to inform/present the team that conducted the tests, as well as the location of the study;
3. This section still lacks important information, such as: inclusion and exclusion criteria (this is a major weakness of the study, so far).
Discussion
1. I suggest creating a subsection at the end of the Discussion entitled "Limitations, strengths and future studies". So far, your study does not present a sufficient number of limitations.
1.1 Therefore, considering that the intention of the study is to discuss a knowledge gap, it is essential to provide the Academic Community with information on how to proceed in the future, as well as suggestions for future studies;
1.2 This sentence "Future research should aim to optimize load-specific rest intervals to maximize performance outcomes without compromising the efficiency of testing protocols" could be included in the new section.
1.3 In turn, merely indicating future research is a bit too little for a scientific publication. After reviewing the literature and conducting this study, the authors should be able to provide more information on how to proceed in the future.
Round 2
Reviewer 1 Report
Comments and Suggestions for Authors
The changes made enhance the validity of the results and conclusions, while also providing additional data to the study. Changes to the abstract, introduction, methods, discussion, and conclusion add significant value to the paper. Furthermore, the addition data in Table 4 and figure 2 greatly improves the presentation of the results for the reader.
In my opinion, these changes are sufficient to approve the paper.